# Sex-specific and concentration-dependent influence of Cremophor RH 40 on ampicillin absorption via its effect on intestinal membrane transporters in rats

Heyue Yin‡, Haibin Shao◎, Jing Liu◎, Yujia Qin◎, Wenbin Deng◎ *

Department of Pharmaceutical Sciences (Shenzhen), Sun Yat-sen University, Guangzhou, China

◎ These authors contributed equally to this work.
‡ HY is the 1st set of equal contributor.
* dengwb5@mail.sysu.edu.cn

**Data Availability Statement:** All relevant data are within the manuscript and its Supporting information files.

## Abstract

Pharmaceutical excipients are the basic materials and important components of pharmaceutical preparations, and play an important role in improving the efficacy of drugs and reducing adverse reactions. Therefore, selecting suitable excipients for dosage form is an important step in formulation development. An increasing number of studies have revealed that the traditionally regarded "inert" excipients can, however, influence the bioavailability of drugs. Moreover, these effects on the bioavailability of drugs caused by pharmaceutical excipients may differ in between males and females. In this study, the in situ effect of the widely-used pharmaceutical excipient Cremophor RH 40 spanning from 0.001% to 0.1% on the intestinal absorption of ampicillin in male and female rats using closed-loop models was investigated. Cremophor RH 40 ranging from 0.03% to 0.07% increased the absorption of ampicillin in females, however, was decreased in male rats. The mechanism of such an effect on drug absorption is suggested to be due to the interaction between Cremophor RH 40 and two main membrane transporters P-gp and PepT1. Cremophor RH 40 altered the PepT1 protein content in a sex-dependent manner, showing an increase in female rats but a decrease in males. No modification on the PepT1 mRNA abundance was found with Cremophor RH 40, indicating that the excipient may regulate the protein recruitment of the plasma membrane from the preformed cytoplasm pool to alter the PepT1 function. This influence, however, may differ between males and females. As such, the study herein shows that supposedly inert excipient Cremophor RH 40 can influence membrane fluidity, uptake and efflux transporters in a sex- and concentration-dependent manner. These findings, therefore, highlight the need for sex-specific studies in the application of solubilizing excipients in drug formulation development.

**Funding:** Dr. Wenbin Deng received the awards and this work was supported by the financial support from the Shenzhen Science and Technology Program (Grant No. KQTD20190929173853397), and the National Natural Science Foundation of China (Grant No. 81772449 and 81971081). The funders had no role in study design, data collection and analysis, decision to publish, or preparation of the manuscript.

**Competing interests:** The authors have declared that no competing interests exist.

## Introduction

Oral absorption is a complex process determined by the interplay of physiological and biochemical processes, physicochemical properties of the active pharmaceutical ingredients (APIs), formulation factors and physiological characteristics of the gastrointestinal tract. Intestinal membrane transporters have also been shown to play an essential role in drug absorption [1]. The extraction of chemicals from the intestinal lumen is achieved by passive diffusion which is also accomplished by a variety of uptake transporters. Likewise, there are apical efflux transporters on enterocytes that prevent the entry of chemicals into the systemic circulation and are often responsible for the poor oral bioavailability of pharmaceuticals. To modify drug absorption, therefore, some relative activators and/or inhibitors are co-formulated with the active pharmaceutical ingredient (API).

Emerging research has demonstrated that allegedly 'inert' excipients are capable of modulating the membrane transporters which can consequently result to enhanced or reduced drug absorption. For instance, polyethylene glycol (PEG), hydroxypropyl-beta-cyclodextrin (HPCD), vitamin E TPGS, DM-β-CyD and Cremophor EL cause pharmacological effect and exhibit substantial interactions with a number of intestinal uptake transporters including organic anion transporting polypeptide (OATPs) [2] and efflux transporters P-glycoprotein (P-gp), multidrug resistant-associated protein 2 (Mrp2) [3–7]. However, limited knowledge and understanding is known regarding the sex differences of these interactions.

The sex-specific influence of excipients on drug bioavailability was first recognized following the study on the widely-used solubilizing agent, PEG 400, where low concentrations of PEG 400 were found to increase the absorption of ranitidine in men but not in women [8]. Interestingly, the sex-related influence of PEG 400 only occurred for drugs whose absorption is controlled by the P-gp efflux transporter such ranitidine and ampicillin in an animal model established for this investigation [9]. It is highly suggested that the sex-based differences in the interaction between PEG 400 and P-gp demonstrates a greater influence in male rats when compared with females [10, 11]. In addition to PEG 400, a portfolio of solubilizing agents such as Cremophor RH 40, Poloxamer 188 and Tween 80 have exhibited a sex-specific reduction on the activity and expression of efflux transporter P-gp to increase the permeability and absorption of P-gp-mediated drug ranitidine [12]. Specifically, Cremophor RH 40 showed the opposite effect on ranitidine intestinal permeability in male and female rats via the Using chamber system [12]. Cremophor RH 40 (macrogolglycerol hydroxystearate polyoxyl 40 hydrogenated castor oil) is a condensation of hydrogenated castor oil with ethylene oxide, which is soluble in most organic solvents, water is dispersed, with excellent emulsification, diffusion properties. Therefore, it has been widely used as a non-ionic solubilizer and emulsifying agent in formulation development.

Moreover, in a previous study on Caco-2 cell lines, the inhibitory influence of Cremophor RH 40 ranging from 0.001% to 3% on P-gp enhanced with the increase of its concentration, with 100% blocking of P-gp by 3% Cremophor RH 40 [13]. Based on this, the effect of excipients on transporters are not only sex-dependent but also dose-dependent.

On the other hand, the underlying mechanism behind this sex-based phenomenon is unclear although a number of hypothesis are possible. The inhibitory on P-gp function could be attributed to downregulation of P-gp expression, inhibition of P-gp ATPase, depletion of intracellular ATP, changes on related nuclear receptors or alteration in membrane fluidity. Thus, to further investigate the reason behind the sex-dependent effect of excipients on the P-gp, the modification on these factors (especially intestinal membrane fluidity) with Cremophor RH 40 in different concentrations will be studied in this work. To determine whether Cremophor RH 40 could alter the function of other main intestinal membrane transporter, such as

oligopeptide transporter 1 (PepT1) and Mrp2, the influence of Cremophor RH 40 on the absorption of ampicillin (the substrate of PepT1, P-gp and Mrp2) was also examined using an in situ closed-loop animal model. Finally, the protein and mRNA expressions of these three transporters (PepT1, P-gp and Mrp2) were quantitated in male and female rats with/without the addition of Cremophor RH 40.

## Materials and methods

### Reagents and materials

Ampicillin sodium was purchased from Bide Pharmatech Ltd. (Shanghai, China). Cremophor RH 40 was purchased from Meilun Biotechnology Company Ltd. (Dalian, China). HEPES, Tris, and EGTA, mannitol, glucose were supplied by Sigma-Aldrich. (USA). Tma-DPH was supplied by BestBio. (Shanghai, China). Water and acetonitrile were purchased from Fisher Scientific (Loughborough, UK) and were of HPLC grade. All other chemicals and kits are noted individually in the following methods.

### Preparation and solubility measurement of solutions

Ampicillin sodium was dissolved in phosphate buffered saline (PBS) at pH 7.4 to yield a final dose of 5 mg/kg in the absence or presence of Cremophor RH 40 (0.001%–0.1%, w/v).

In order to determine the saturated solubility of ampicillin in different Cremophor RH 40 solutions, samples were prepared by adding an excess amount of compound to different Cremophor RH 40 solutions in vials of 10 mL. The vials were kept under stirring at 100 rpm and controlled temperature at 37°C (±0.1°C). The solubility of ranitidine in the different solutions was: 0 g Cremophor RH 40 (410 mg/ml); 0.001% Cremophor RH 40 (417 mg/ml); 0.01% Cremophor RH 40 (409 mg/ml); 0.03% Cremophor RH 40 (400 mg/ml); 0.05% Cremophor RH 40 (421 mg/ml); 0.07% Cremophor RH 40 (424 mg/ml); 0.1% Cremophor RH 40 (414 mg/ml).

### Animals

Animals used for the experiments were male and female Wistar rats (8 weeks old, 24 males and 24 female rats in total) purchased from the Southern Medical University Laboratory Animal Centre (Guangzhou, China). Male and female Wistar rats weighed approximately 300 g and 250 g respectively. The animal protocol was approved by the Sun Yat-sen University Animal Use and Care Committee (SYSU-IACUC-2021-000319).

### Influence of cremophor RH 40 on ampicillin absorption

*In situ* closed-loop model set-up.    Animal work for screening the potential sex-effect of Cremophor RH 40 from 0.001% to 0.1% concentrations were performed at the Sun Yat-sen University Laboratory Animal Centre. The protocol was agreed by the Administrative Committee on Animal Research in Sun Yat-sen University. All rats were housed at room temperature (25°C) and in a light-dark cycle of 12 h. The rats were caged in groups of 4, allowed to move freely and provided with food and water before the experiment. The day before the experiment, the rats were fasted overnight and individually housed in metabolic cages.

On the day of the experiment, each rat was weighed and anesthetized with Somnopentyl® (sodium pentobarbital, 32 mg/kg body weight intraperitoneally). The intestine was exposed through midline abdominal incision. After ligating the bile duct, a segment of jejunum (about 20 cm long) was isolated and washed with PBS and then tied off at both ends to form a closed loop. The drug solution was warmed to 37°C and 3 mL was injected into the jejunum loop. Subsequently, approximately 250 μL–300 μL of blood was collected from the rats' tail vein and

transferred to anticoagulant centrifuge tubes (BD Microtainer® K2E Becton, Dickinson and Company, USA) at predetermined time intervals up to 240 min. At the end of the experiment, the luminal solution was collected to measure the remaining ampicillin, the amount of protein released and the activity of lactate dehydrogenase (LDH). Also, the jejunum loop tissues were kept at -80˚C until use.

**Preparation of blood samples.** Blood samples were centrifuged at 10,000 rpm for 10 min and the supernatants (plasma samples) were collected into 1.5 mL Eppendorf tubes, the same volume of methanol was added to precipitate the plasma proteins. After 1 min of vortex-mixing, the mixture was centrifuged at 4˚C for 10 min at 10,000 rpm. The supernatant was collected and 50 μL aqueous was analysed by HPLC.

**HPLC analysis.** Chromatographic analysis was performed with a HPLC system (Waters Breeze) equipped with Binary HPLC pump (Waters 1525), autosampler (Waters 2707) and an UV/Visible detector (Waters 2489). The drug was quantified by HPLC using a Luna C18 (250 mm × 4.6 mm I.D./5 μm) column (FLM, Guangzhou, China) using 10 mM sodium dihydrogen phosphate buffer (pH 7.0)- methanol (60:40, v/v). The flow rate was 1 mL/min, and the UV detector was set at 220 nm.

**Pharmacokinetic analysis.** Pharmacokinetic parameters ($C_{max}$, $t_{max}$, $AUC_{0-240}$ and $AUC_{\infty}$) were calculated by non-compartmental analyses using a free Microsoft Excel add-in "PKSolver."

## Effect of cremophor RH 40 on intestinal membrane damage

The release of protein and the activity of LDH in small intestinal membranes were used to evaluate small intestinal membrane damage in the presence of Cremophor RH 40 using an *in situ* closed-loop method. Triton X-100 (3%, v/v) were selected as the positive control in this study by being administered into the intestinal loop. The collected luminal solution was washed with PBS for the determination of the amount of protein released and the activity of LDH. The concentrations of protein released from the intestinal membranes were measured with BSA as a standard using the BCA Assay Protein kit (Beyotime Biotechnology, Shanghai, China). The activities of LDH were determined using the LDH CII Assay Kit according to the manufacturer's instructions.

## Measurement of PepT1, P-gp and Mrp2 protein levels in rat intestine by ELISA

The in situ jejunal loop after experiments was excised and washed with cold PBS. To obtain the mucosal tissue, 5 cm tissues were placed on an ice-cold glass plate and the serosa layer was gently squeezed out with a cover slip, following divided into aliquots for determination of P-gp protein content.

The mucosal tissues (about 50 mg) of the intestinal of rats were cut into small pieces and homogenized in 0.5 ml RIPA lysis buffer at 30 Hz for 30 s with a TissueLyser (QIAGEN, Hilden, Germany), and repeated twice at intervals of 30s to completely homogenized. The tissue homogenates were incubated at 4˚C for 2h, then centrifuged at 4˚C, 12,000 g for 5 min. The total tissue protein was collected in the supernatants, and its concentration was subsequently determined with the BCA Assay Protein kit (Beyotime Biotechnology, Shanghai, China) according to the manufacturer's instructions. To measure the targeted transporter protein levels, PepT1, P-gp and MRP2 was quantified by ELISA kits (Meimian Biotech, Yancheng, China) based on the manufacturer's description.

## Measurement of PepT1, P-gp and Mrp2 mRNA expression in intestinal segments by real time reverse-transcription polymerase chain reaction (RT-PCR)

Total RNA in each intestinal sample was isolated and purified with Tissue RNA Purification Kit Plus (ESscience Biotech, Shanghai, China), and RNA concentration was measured with Nanodrop 2000 (Thermofisher) according to the manufacturer's instructions.

Subsequently, the quantification of the target RNA was conducted as follows; 0.5 µg total RNA of each sample was reverse transcribed using the Fast Reverse Transcription Master Mix (Aivd Biotech, Shenzhen, China). To quantify the amount of PepT1 mRNA, P-gp mRNA (mdr1a and mdr1b) and mrp2 mRNA, RT-PCR was performed on the LightCycler® 96 (Roche, Basel, Switzerland) using the method described in MacLean's study [14]. Briefly, 20 µL PCR reaction contained 10 µL of AB-HS SYBR Green qPCR Mix (Aivd Biotech), 2 µL each of forward and reverse primers, and 2 µL of cDNA, and 4 µL ddH2O. RT-PCR was carried out in 96 well PCR plates (Thermofisher). The amplification program for all genes consisted of one pre-incubation cycle at 95˚C with a 2 min hold, followed by 45 amplification cycles with denaturation at 95˚C with a 10s hold, an annealing temperature of 60˚C with a 20 s hold, and an extension at 72˚C with a 15 s hold. Amplification was followed by a melting curve analysis, which ran for one cycle with denaturation at 95˚C with a 1 s hold, annealing at 65˚C with a 15 s hold, and melting at 95˚C with a 1 s hold. Distilled water was included as a negative control in each run to access specificity of primers and possible contaminants.

Primers (shown in Table 1) were designed by primer-BLAST searching with publicly available sequence information of the Gene Bank of the National Center for Biotechnology Information (NCBI) and purchased from TSINGKE (Beijing, China). Relative expression of PepT1, *mdr1a*, *mdr1b* and mrp2 mRNA in the intestinal was calculated using LightCycler® 96 software (version 1.1, Roche). The average of the threshold cycle (Ct) values for tested genes (PepT1, *mdr1a*, *mdr1b* and mrp2) and the internal control (anti-beta actin) was taken, and then the differences between Ct values for tested genes and internal control (ΔCt) were calculated for all the experimental samples.

## Effect of cremophor RH 40 on intestinal membrane fluidity

**Preparation of the rat intestinal brush border membrane vesicles.**   Brush border membrane vesicles (BBMVs) were prepared by the divalent cation precipitation method using MgCl$_2$ in the presence of ethylenebis (oxyethylenenitrilo) tetraacetic acid (EGTA), as described previously. In brief, an 5 cm in situ small intestinal loop in male and female Wistar rats was excised and divided into segments, and the mucosa was scraped with a slide glass and homogenized in 10 volumes of buffer (5 mM EGTA, 12 mM Tris, 300 mM mannitol; pH 7.4 by 1 M 2-

**Table 1.  Primers used for the analysis of P-gp and PepT1 gene expression by RT-PCR.**

| Gene | | Primer (5′– 3′) | Amplicon (bp) |
|---|---|---|---|
| *mdr1a* | Forward | 5′-CACCA TCCAGAACGCAGACT-3′ | 139 |
| | Reverse | 5′-ACATCTCGCATGGTCACAGTT-3′ | |
| *mdr1b* | Forward | 5′-AACGCAGACTTGATCGTGGT-3′ | 144 |
| | Reverse | 5′-AGCACCTCAAATACTCCCAGC-3 | |
| *rPept1* | Forward | 5′-GTATGTTCTGTTCGCCTCCTTG-3′ | 228 |
| | Reverse | 5′-GGTGAATGCTGGACTTGGTATG-3′ | |
| anti-beta actin | Forward | 5′-GCAGGAGTACGATGAGTCCG-3′ | 74 |
| | Reverse | 5′-ACGCAGCTCAGTAACAGTCC-3′ | |

[4-(2-hydroxyethyl)piperazin-1-yl]ethanesulfonic acid (HEPES)) with a TissueLyser (QIA-GEN, Hilden, Germany). A stock solution of 1 M $MgCl_2$ was added to the homogenate to give a final concentration of 10 mM $MgCl_2$. The mixture was gently stirred for 1 min and allowed to stand at $4^oC$ for 15 min. Then, the suspension was centrifuged at 3000 g for 15 min. The pellets were discarded, and the supernatant was centrifuged again at 32,000 g for 30 min. The pellets containing BBMVs were resuspended by 27-gauge needle in the homogenizing buffer. The protein concentrations were determined by BCA Assay Protein kit (Beyotime Biotechnology, Shanghai, China) using BSA as a standard, and the final concentration was adjusted to 1 mg/mL in each tube. The samples were frozen by liquid $N_2$ and kept at $-80^oC$ until use.

**Measurement of membrane fluidity by fluorescence polarization.** Fluorescence polarization techniques were used to evaluate the effects of Cremophor RH 40 on small intestinal membrane fluidity using the fluorescence probes Tma-DPH. BBMV suspensions (protein concentration, 100 mg/mL) were incubated with 0.5 μM Tma-DPH for 5 min in Tris-HEPES buffer (25 mM HEPES, 5 mM glucose, 140 mM NaCl, 5.4 mM KCl, 1.8 mM CaCl2, 0.8 mM MgSO4; pH 7.4 with 1 M Tris). 200 mM cholesterol as a positive control was added. Then, the fluorescence intensities and steady-state polarization were determined using a Hitachi F2000 spectrophotometer (Hitachi, Yokohama, Japan) equipped with a polarizer set. The fluorescence polarization ($\Im$) was calculated following Eq 1:

$$\Im = (I_{\parallel} - G \cdot I_{\perp})/(I_{\parallel} + G \cdot 2 \cdot I_{\perp}) \tag{1}$$

where $I_{\parallel}$ and $I_{\perp}$ represent the fluorescent intensities of the parallel and perpendicular polarized excitation, respectively. G is the equipment parameter. The excitation and emission wavelengths were 360 nm and 430 nm, respectively, for Tma-DPH.

## Statistical analysis

All results are expressed as mean ± SD (n = 4). The control and test group data were analysed by one-way ANOVA followed by post-hoc Tukey analysis with a 95% confidence interval using IBM SPSS Statistics 16 (SPSS Inc., Illinois, USA).

## Results

The intestinal absorption of 5mg/kg ampicillin in the absence of excipients presented by the cumulative area under the plasma concentration versus time curve ($AUC_{0-240min}$) in male and female rats was 4693 ± 1652 μg.min/mL and 786 ± 189 μg.min/mL respectively, described in Fig 1a. As shown in Fig 1b, the plasma concentrations of ampicillin were markedly different in male and female rats. And the effect of Cremophor RH 40 on the pharmacokinetic parameters of ampicillin in male and female Wistar rats is shown in S1 Table. In terms of the membrane transporters, significant higher PepT1, Mrp2 and P-gp protein levels were found in male rats compared with the female groups (shown in Fig 1c, 1e and 1g). P-gp related *mdr1a* mRNA expression also showed a sex-specific difference indicating higher abundance in males (Fig 1d), while *mdr1b* contents were similar in both sexes. As found in Fig 1f and 1h, Mrp2 and rPept1 mRNA expression was consistent with its protein levels, revealing a slightly higher content in the jejunum of male group.

Assays for LDH and the release of protein were conducted to measure membrane damage by Cremophor RH 40 in the small intestine by an *in situ* closed-loop experiment. The results showed the activity of LDH and the amount of protein released were not significantly different compared with the control.

The presence of 0.03% to 0.07% Cremophor RH 40 excipient concentrations significantly increased the intestinal absorption of ampicillin in female rats by 41%, 32% and 25%

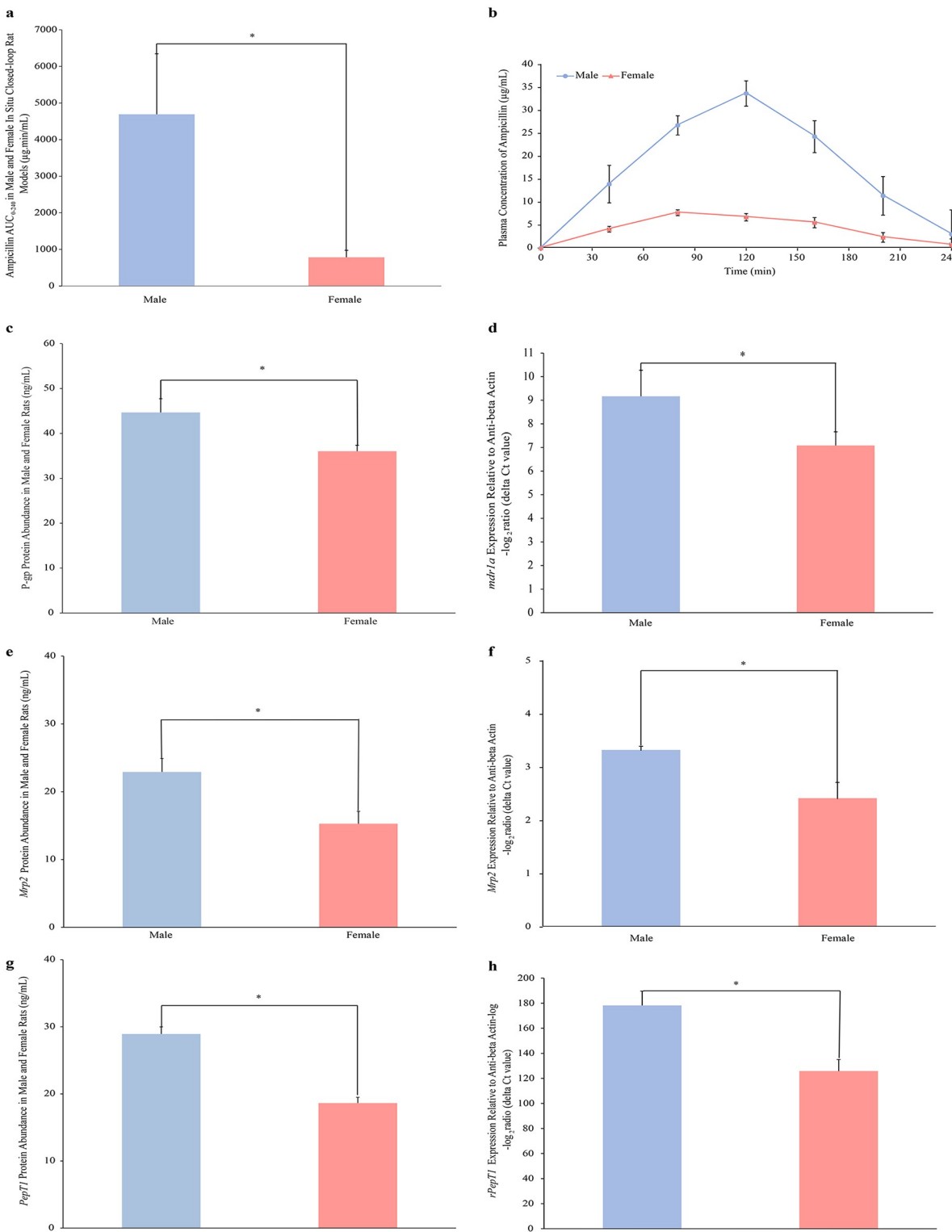

**Fig 1. Related pharmacokinetic parameters and intestinal membrane transporter expression in male and female Wistar rats.** (a) the absorption (AUC0–240min) of 5mg/kg ampicillin; (b) plasma concentration-time profile of ampicillin; (c) P-gp protein abundance; (d) mdr1a mRNA expression; (e) Mrp2 protein content; (f) Mrp2 mRNA level; (g) PepT1 protein expression; (h) rPepT1 level. All the data were expressed as mean ± standard error of at least 4 experiments. * Values are statistically different between the male and female groups.

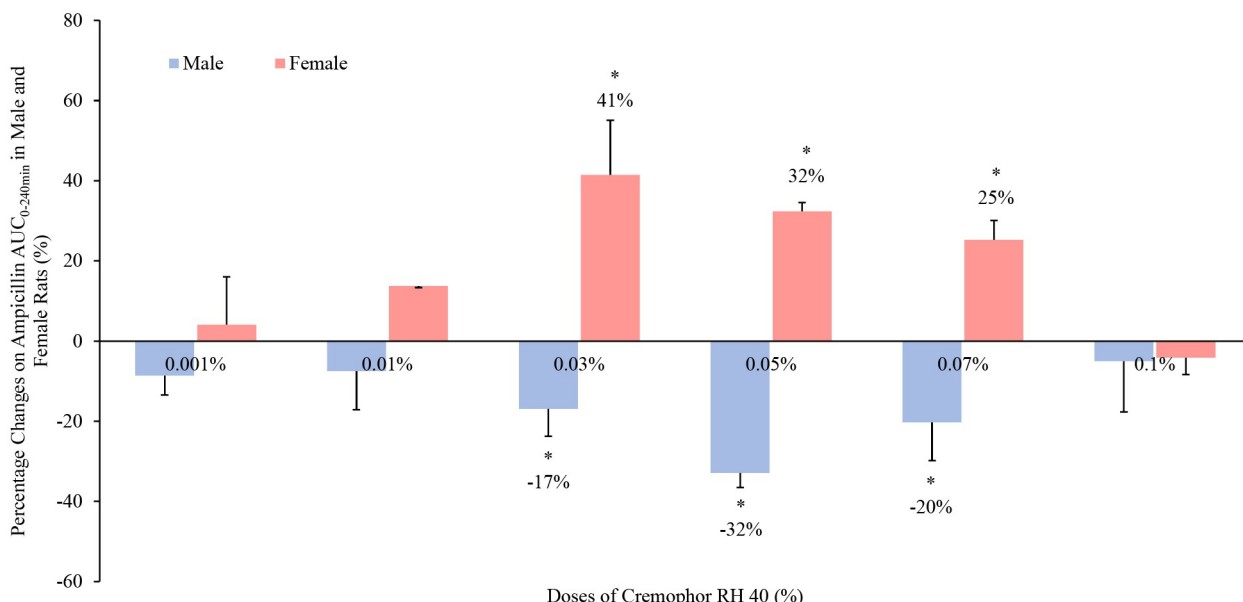

**Fig 2. Percentage change in the absorption (AUC0–240min) of ampicillin (5 mg/kg) in the presence of Cremophor RH 40 in the jejunum of male and female Wistar rats by an in situ closed-loop study (mean ± SD, n = 4).** * Values are statistically different between the control (i.e. no Cremophor RH 40) and the Cremophor RH 40 groups at $p < 0.05$.

respectively ($p < 0.05$). However, in their male counterparts, ampicillin bioavailability reductions were observed with Cremophor RH 40 in these doses (0.03%, 0.05% and 0.07%) indicating a 17%, 32% and 20% decrease over the control ($p < 0.05$) (Fig 2).

With regards to the P-gp protein and gene abundance, Cremophor RH 40 ranging from 0.001% to 0.05% had no influences on the P-gp levels (protein and mdr1a) in neither male rats nor females. However, 0.07% and 0.1% Cremophor RH 40 decreased its expression in males, while no alteration was observed in their female counterparts (Figs 3 and 4). It can also be seen from S1 Fig, the intestinal mdr1b expression in both males and females fluctuated with Cremophor Rh 40 dose, and no trend could be observed. Thus, a strong positive correlation between mdr1a (but not mdr1b) expression and P-gp protein abundance, with high levels of P-gp protein associated with high levels of mdr1a expression.

From S2 Fig, the Mrp2 protein expression was not altered with the addition of Cremophor RH 40 in different concentrations in neither male nor female rats ($p > 0.05$), which also indicated no sex differences on the effect of excipients on the function of Mrp2.

In relation to the uptake transporter Pept1 (obtained in Fig 5), the presence of 0.03% to 0.07% concentrations of Cremophor RH 40 enhanced the protein abundance by 48%, 36% and 23% in female rats. Although an enhancement in PepT1 protein level was achieved with 0.1% Cremophor RH 40 in female rats, this result was not statistically significant ($p > 0.05$). However, Cremophor RH 40 reduced the protein content in male rats, the greatest reduction of 29% was demonstrated in the presence of 0.1% Cremophor RH 40 when compared with the control ($p < 0.05$). Moreover, the presence of Cremophor RH 40 was not able to alter the Pept1 mRNA expression in both male and female rats, and this was not a clear trend (present in Fig 6).

To elucidate the mechanisms of transporter alteration by Cremophor RH 40 in different concentrations, intestinal membrane fluidity was measured by fluorescence polarization. The polarization of Tma-DPH fluorescence probes was significantly decreased in males but not in

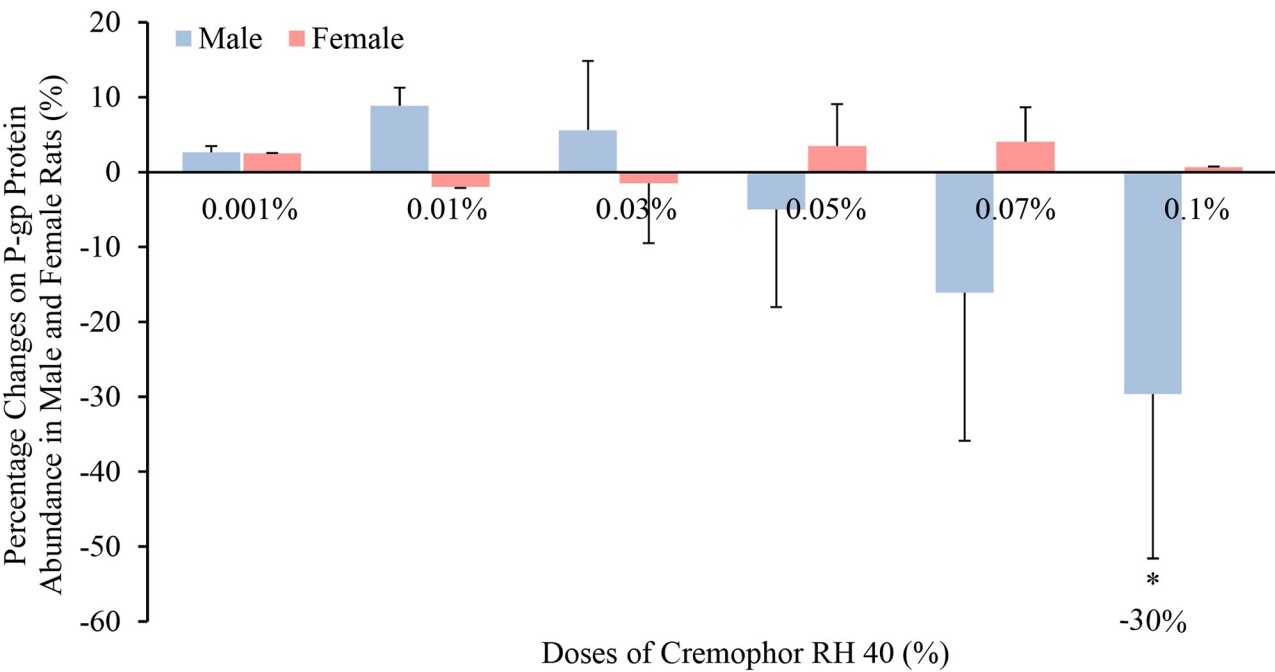

**Fig 3. Percentage change in P-gp protein level in the presence of Cremophor RH 40 in the jejunum of male and female Wistar rats (mean ± SD, n = 4).** * Values are statistically different between the control (i.e. no Cremophor RH 40) and the Cremophor RH 40 groups at p < 0.05.

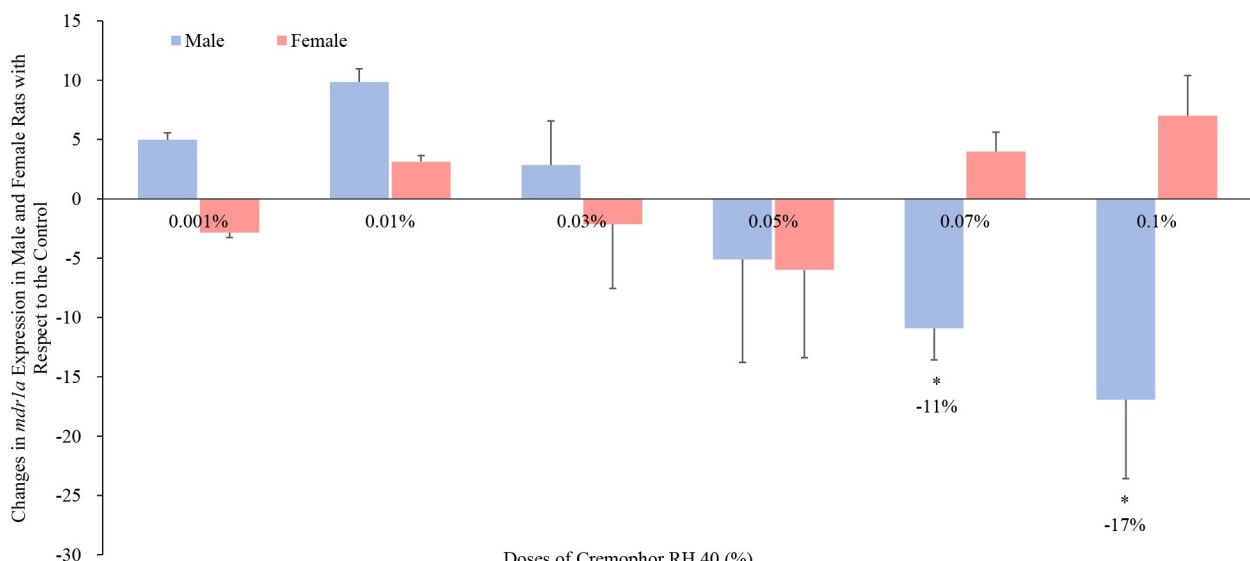

**Fig 4. Percentage change in mdr1a mRNA expression in the presence of Cremophor RH 40 in the jejunum of male and female Wistar rats (mean ± SD, n = 4).** * Values are statistically different between the control (i.e. no Cremophor RH 40) and the Cremophor RH 40 groups at p < 0.05.

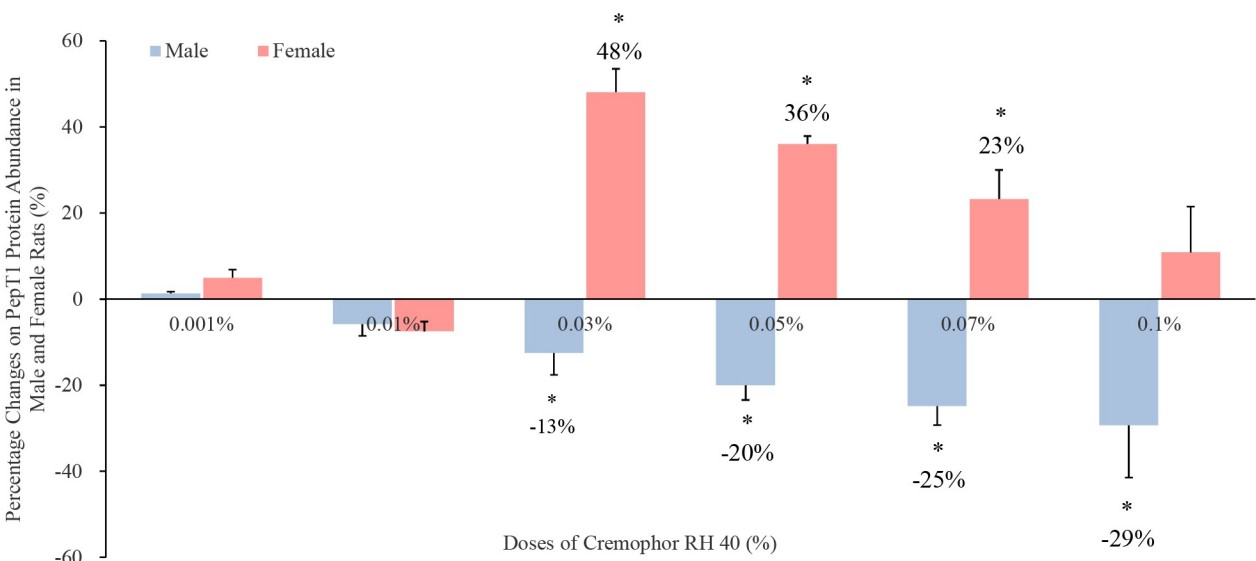

**Fig 5. Percentage change in PepT1 protein level in the presence of Cremophor RH 40 in the jejunum of male and female Wistar rats (mean ± SD, n = 4).** * Values are statistically different between the control (i.e. no Cremophor RH 40) and the Cremophor RH 40 groups at $p < 0.05$.

female rats by the addition of higher concentrations (0.07% and 0.1%, w/v) of Cremophor RH 40, compared with that of control group (shown in Fig 7).

## Discussion

The values of ampicillin *in vivo* bioavailability were not consistent with the results reported in previous study [9] where drug absorption was higher in female rats, compared with the males. The intestinal absorption of ampicillin obtained using *in situ* closed-loop rat model in the present study was, however, 5-fold higher in male rats than females. This may be explained by the

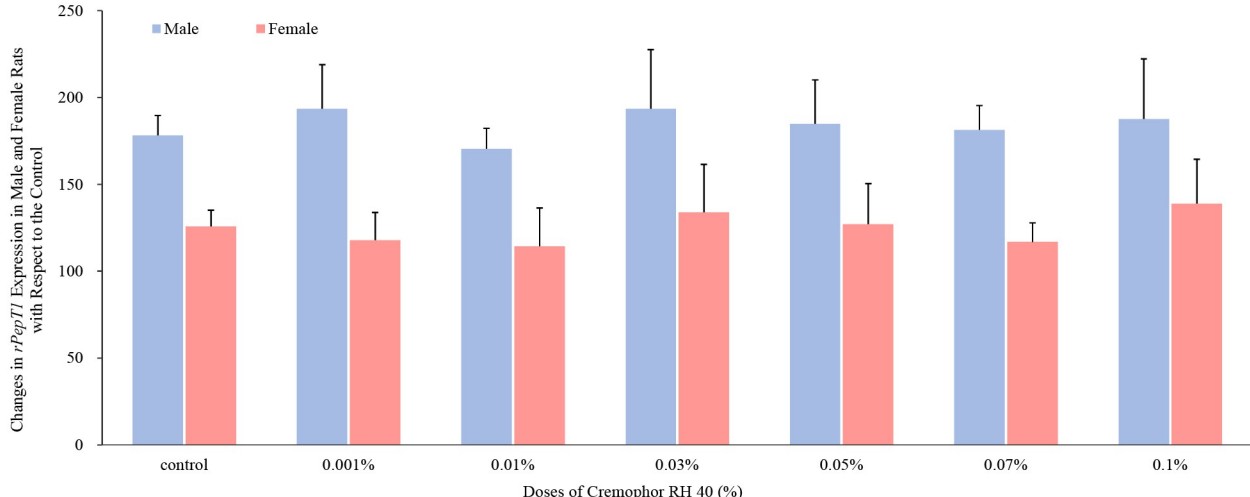

**Fig 6. Percentage change in rPepT1 mRNA expression in the presence of Cremophor RH 40 in the jejunum of male and female Wistar rats (mean ± SD, n = 4).** No significant differences were observed ($p > 0.05$).

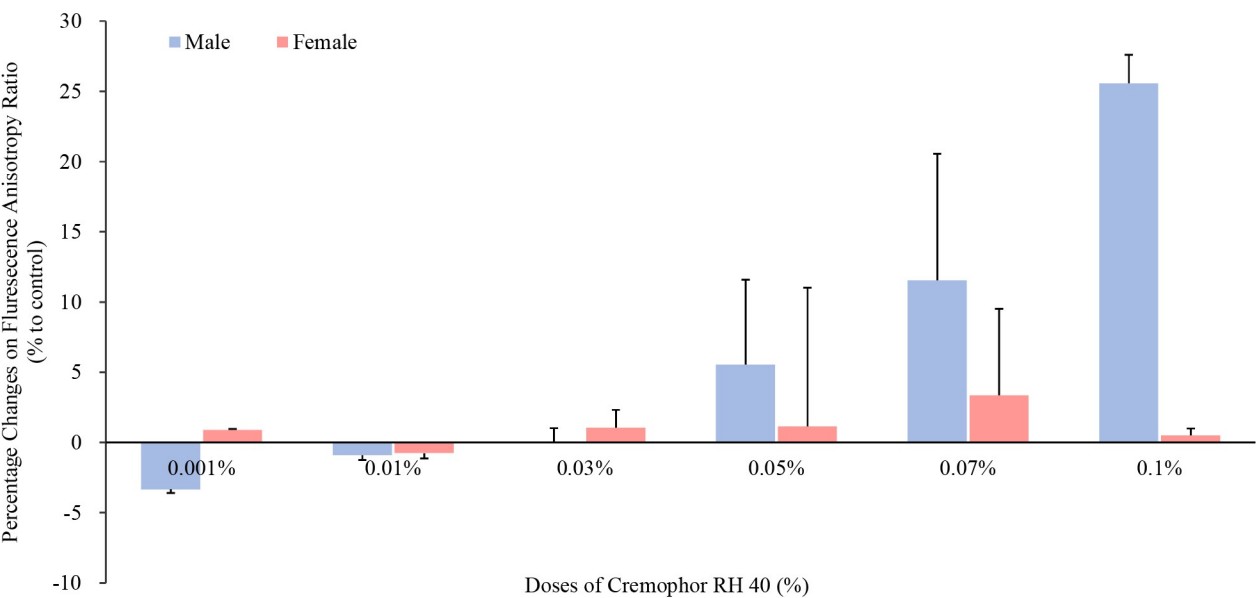

**Fig 7. Influences of Cremophor RH 40 ranging from 0.001% to 0.1% on BBMVs intestinal membrane fluidity of the male and female Wistar rats (mean ± SD, n = 12), present by fluorescence anisotropy of the intestinal membrane labeled with Tma-DPH as a probe.** * Values are statistically different between the control (i.e. no Cremophor RH 40) and the Cremophor RH 40 groups at p < 0.05.

different ampicillin dosages, routes of administration, experimental models and the period of pharmacokinetic studies. To be specific, the transit time differed between males and females which has been excluded in the *in situ* closed-loop rat model study [15]. Also, the exposure areas of intestine to drug solution was much larger in the *in vivo* rat study than that during the *in situ* mode. All these differences between these two experiments might lead to the variations.

As the substrate of intestinal membrane transporter PepT1, P-gp and Mrp2, ampicillin absorption was related to the abundance of these proteins [16–18]. There were a few studies that reported sex-related differences in the expression of uptake transporter PepT1 in the intestine, PepT1 mRNA was once reportedly highly expressed in small intestine compared with other organs and no significant difference between sexes in Sprague-Dawley rats and C57BL/6 mice [19]. However, based on the results herein, female Wistar rats demonstrated a higher PepT1 regardless of protein and gene expression, which also well supported the *in situ* intestinal absorption data. On the other hand, higher P-gp expression in male rats than females was in agreement with values reported in the previous literature [10, 11]. Surprisingly, higher P-gp should result in lower ampicillin absorption in males compared with female rats, the opposite findings would be attributed to the larger influences of intestinal PepT1 levels, which suggested the oral absorption of ampicillin was mainly influenced by the uptake transporter, PepT1, in both male and female rats. In terms of Mrp2, our findings were also fully supported by an earlier work where demonstrated no significant differences in Mrp2 protein and mRNA expression between males and females in the rodents' intestine [14].

The alteration on ampicillin intestinal absorption with excipients, however, was demonstrated in a concentration- and sex-dependent manner. Although the bioavailability of a drug is influenced by many factors, the solubility of ampicillin was specifically measured and Cremophor RH 40 from 0.01% to 0.1% had no statistically significant effect (p > 0.05) on the solubility of ampicillin, thereby eliminating it as a reason for any observed increases in drug absorption.

Cremophor RH 40 is frequently used as a solubilizing enhancer in the oral formulations. Recent research, however, have reported its inhibitory effect on P-gp to enhance the drug permeability at a dose higher than its critical micelle concentration (CMC) [12]. In this study, the modification on ampicillin absorption by Cremophor RH 40 was, however, resulted from the sex-dependent influences on two membrane transporters mentioned above, P-gp and PepT1. No significant alteration was observed on the efflux transporter Mrp2 by the addition of Cremophor RH 40.

On the one hand, regarding the P-gp efflux transporter, our results demonstrate the influence of Cremophor RH 40 at lower doses showed no effects on P-gp protein levels in rats while higher concentrations (0.07% and 0.1%) displayed a P-gp inhibitory, reflecting similar findings to previous investigations; in a study by Wandel *et al.*, the effects of low concentrations of Cremophor RH 40 (0.001% to 3%) as potential P-gp inhibitors were tested with the use of digoxin transport in Caco-2 cells. The results showed that the increase in Cremophor RH 40 concentration inhibited P-gp efflux transporter ability at a greater extent and achieving complete P-gp inhibition following the administration of 3% Cremophor RH 40 [13]. However, there is limited knowledge and understanding in the effect of Cremophor RH 40 on P-gp expression in females. A clear statistical contrast to that seen in male rats ($p < 0.05$) was observed in our study. Specifically, in females, the intestinal P-gp abundance was not influenced in the presence of Cremophor RH 40 in all the tested concentrations (ranging from 0.001% to 0.1%). The reason behind this sex-related influence of excipients on efflux transporter P-gp was yet unknown.

Alternatively, it is possible that Cremophor RH 40 may cause the intestinal membrane damage and enhance intestinal absorption of P-gp substrates. However, Cremophor RH 40 in all tested doses had no effect on the activities of LDH and the amount of protein compared to the control group. Thus, Cremophor RH 40 at a low concentration could be considered to be safe P-gp modulators to increase the intestinal absorption of P-gp substrates via reducing the P-gp expression in males. Moreover, P-gp is reportedly sensitive to the lipid environment and might be involved in lipid trafficking and metabolism [20]. Our findings indicated that Cremophor RH 40 at 0.07% and 0.1% concentrations might increase the membrane fluidity of lipid bilayers and the protein portion of the membrane in male rats but not female, especially the fluidity of the hydrophobic core of the outer lipid bilayers, because the maximal reducing effect was observed in the fluorescence polarization of Tma-DPH, which probes the fluidity of the polar head group region of the outer lipid bilayer [21]. This also, in the first time, suggested the sex-dependent influence of 'inert' pharmaceutical excipients on the intestine membrane fluidity.

On the other hand, the expression and function of the influx transporter PepT1 can be modulated by numerous factor, such as substrates, proteins, hormones and diseases status [22]. In present study, intestinal PepT1 protein expression was first reported to be regulated by excipient Cremophor RH 40, even in a dose- and sex-based manner. Based on the previous literatures, alteration on the function of PepT1 protein were conducted via two mechanisms. One is a long-term modification, which is influenced by the transcriptional and/or post-transcriptional status. For example, 1 day of fasting could increase the population of Pept-1 in the brush-border membrane of rat intestine via mediation on peroxisome proliferator-activated receptor α (PPARα) to induce dipeptide transport, which was considered in the transcriptional level [23, 24]. The post-transcriptional regulation of PepT1 caused by the postsynaptic density-95/disk-large/ZO-1 [PDZ] domain-containing protein (PDZK1) was first identified in 2008 [25]. In this study, a significant decrease in the PepT1's location at the apical membrane of small intestinal epithelial cells was found in PDZK1 gene knockout mice. However, insulin and leptin were involved in the regulation of PepT1 even with different molecular mechanism,

which were short-term action. Thamotharan *et al*. hardly ever discovered changes in the PepT1 mRNA level in Caco-2 cells incubating with insulin, while the protein level of PepT1 in the apical membrane was increased [26, 27]. Following this, they proposed that insulin may regulate PepT1 via increasing the protein recruitment of the plasma membrane from the pre-formed cytoplasm pool. That is to say, although insulin cannot be found in the gut lumen, the peptide transport activity still can be regulated by binding to its receptor located in the basolateral membrane of the intestinal mucosal cells [28]. Moreover, leptin increased the membrane PepT1 protein, decreased intracellular PepT1 content, however, showed no effects in rPepT1 mRNA levels [29]. The findings in our study indicating that Cremophor RH 40 regulated the PepT1 protein abundance but not mRNA level was similar to the insulin- and leptin-induced work. Therefore, Cremophor RH 40 may modify the protein recruitment of the plasma membrane from the preformed cytoplasm pool to alter the PepT1 function, and this modification might differ between males and females.

To our knowledge, mammals absorb various substrates in the gastrointestinal tract through PepT1, the regulation of PepT1 has implications in studies of enteral nutrition and especially drug therapy [22]. The sex-dependent effect of pharmaceutical excipients Cremophor RH 40 on the PepT1 function, for the first time, was reported in this study. The mechanism behind this sex-specific phenomenon was still unknown. However, the usage of personalized cell model (such as intestinal organoid, intestine-on-a-chip) [30–32], in-depth study of the 3D structure and factors affecting PepT1 activity and mechanisms of action can better provide information to investigate the reason underlying this sex-based influence of excipients on the PepT1 expression.

Great progress has been made in the application of excipients in formulation development, not only for predetermined purposes but to actively modulate drug bioavailability according to the work reported here. The co-formulation of pharmaceutically relevant excipient concentrations of Cremophor RH 40 was able to actively modulate the enhancement of ampicillin permeability and intestinal PepT1 expression, however, this was limited to male and not female rats. The sex differences demonstrated by this study highlights the need for the screening of "inert" pharmaceutical excipients in terms of their selection and use for oral drug development, particularly for PepT1 substrates, in order to negate potential adverse events from an increase in bioavailability or from the lack of therapeutic effect. Also, the application of these sex-specific excipients in oral formulation developments could lead to a 'sex-related' formulation and/or personalized formulation field, resulting in better therapeutic efficacy and lower side effects, and supporting precision medicine.

## Supporting information

**S1 Fig. Percentage change in *mdr1b* mRNA expression in the presence of cremophor RH 40 in the jejunum of male and female Wistar rats (mean ± SD, n = 4).**
(TIF)

**S2 Fig. *Mrp2* protein level in the presence of cremophor RH 40 in the jejunum of male and female Wistar rats measured by ELISA (mean ± SD, n = 4).**
(TIF)

**S1 Table. Effect of Cremophor RH 40 on the pharmacokinetic parameters of ampicillin in male and female Wistar rats (mean±standard deviation, n = 4).**
(DOCX)

**S1 Graphical abstract.**
(JPEG)

## Acknowledgments

We are grateful to Dr. Mai Y. for her advice and guidance on the subject. We also thank the scientific research platform provided by Sun Yat-Sen University for the smooth progress of our experiments.

## Author Contributions

**Conceptualization:** Heyue Yin, Haibin Shao.

**Data curation:** Heyue Yin.

**Formal analysis:** Heyue Yin.

**Investigation:** Haibin Shao.

**Methodology:** Heyue Yin.

**Project administration:** Wenbin Deng.

**Resources:** Haibin Shao.

**Software:** Haibin Shao.

**Supervision:** Wenbin Deng.

**Validation:** Heyue Yin, Jing Liu, Yujia Qin.

**Visualization:** Wenbin Deng.

**Writing – original draft:** Heyue Yin.

**Writing – review & editing:** Wenbin Deng.

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
