## [Decision Letter · Decision Letter 0]

22 Dec 2021

PONE-D-21-36367Sex-specific and concentration-dependent influence of Cremophor RH 40 on ampicillin absorption via its effect on intestinal membrane transportersPLOS ONE

Dear Dr. Deng,

Thank you for submitting your manuscript to PLOS ONE. After careful consideration, we feel that it has merit but does not fully meet PLOS ONE’s publication criteria as it currently stands. Therefore, we invite you to submit a revised version of the manuscript that addresses the points raised during the review process.

We look forward to receiving your revised manuscript.

Kind regards,

Chun-Hua Wang

Academic Editor

PLOS ONE

Journal Requirements:

"We are grateful to Dr. Yang Mai for her advice and guidance on the subject.. We also thank the scientific research platform provided by Sun Yat-Sen University for the smooth progress of our experiments. This work was supported by the financial support from the Shenzhen Science and Technology Program (Grant No. KQTD20190929173853397), and the National Natural Science Foundation of China (Grant No. 81772449 and 81971081)."

"Dr. Wenbin Deng received the awards and this work was supported by the financial support from the Shenzhen Science and Technology Program (Grant No. KQTD20190929173853397), and the National Natural Science Foundation of China (Grant No. 81772449 and 81971081).

5. Please include a copy of Table 1 which you refer to in your text on page 10.

Reviewers' comments:

Reviewer's Responses to Questions

**Comments to the Author**

1. Is the manuscript technically sound, and do the data support the conclusions?

Reviewer #1: Yes

Reviewer #2: Yes

Reviewer #3: Yes

Reviewer #4: Yes

Reviewer #5: Yes

2. Has the statistical analysis been performed appropriately and rigorously? 

Reviewer #1: Yes

Reviewer #2: Yes

Reviewer #3: Yes

Reviewer #4: I Don't Know

Reviewer #5: Yes

3. Have the authors made all data underlying the findings in their manuscript fully available?

Reviewer #1: Yes

Reviewer #2: Yes

Reviewer #3: No

Reviewer #4: Yes

Reviewer #5: Yes

4. Is the manuscript presented in an intelligible fashion and written in standard English?

Reviewer #1: Yes

Reviewer #2: Yes

Reviewer #3: Yes

Reviewer #4: Yes

Reviewer #5: Yes

5. Review Comments to the Author

Reviewer #1: The topic is so interesting and the research findings are well presented. Authors had discussed their results appropriately and had selected good references to support it.

The only remark is that the resolution of Figure 1 needs improvement because it is not readable.

Reviewer #2: Line 71-73 Cite a reference number for this statement.

101 Define what is PBS

104-107 Provide the number of male rats and the number of female rats used for the study separately. In the section 2.3.

127 Define what is LHD

143 According to the manuscript pharmacokinetic parameters of Cmax, tmax, AUC0-240 and AUC∞ have been calculated. However, in the results section all these parameters except AUC0-240 have to be deduced by the reader from the Figure 1b. It is required to provide the results for these parameters with the Figure 1b or separately.

145 Describe the positive and negative controls used in determining the effect of cremophor RH 40 on intestinal membrane damage.

164 Remove “a” before at

181 Provide the reference number for MacLean's study here

241-242 It is not clear the shown plasma concentration values are corresponding to what inlet concentration/s of ampicillin. Provide this information both in the paragraph here and in Figure 1b

255-257 Rather than given a range of 0.03% - 0.07%, provide the 3 Cremophor RH 40 concentrations resulting 41%, 32% and 25% increase in ampicillin absorption

310-312 If this hypothesis of possible influence of PepT1 is used to explain ampicillin absorption in male rats, how the same argument to be used to explain same thing in female. Need a discussion with respect to female rats on the same line.

318-321 Solubility measurements done is not mentioned in the methodology and in the results section. Therefore, bringing a new result in the discussion is not acceptable. Include solubility measurement method and results in the relevant sections.

Figures Poor contrasting and clarity. Need improvements

Figure 1 Write the units of the y-axis

Reviewer #3: The authors delivered a considerable work of the sex-specific and concentration-dependent influence of Cremophor RH 40 on ampicillin absorption. This work revealed the mechanisms of Cremophor RH 40 on intestinal membrane transporters to some extent, and supplied some valuable experience to evaluate the use of Cremophor RH 40 in the design and development of oral delivery systems.

However, there are still some aspects to discuss.

1. It seems irrational or inexactly that the dose of Cremophor RH 40 in the experiment was noted as volume in volume, since Cremophor RH 40 as a semi-solid excipient at room temperature, cannot be exactly measured by volume even at heated status. Also the weight dose of ingredients administrated orally was preferred rather than volume.

2. According to the results of AUC and Cmax after oral administration of 50mg/kg ampicillin mentioned in reference 9, the AUC and Cmax herein were 5-fold higher in male rats after 5mg/kg ampicillin injected into jejunum closed-loop (one tenth of the dose in reference 9). Explanations with more details should be available in line 299 of the manuscript.

3. The pharmacokinetic parameters (Cmax, tmax, AUC0-240 and AUC∞) calculated in section 2.4.4, should be listed as a table form in the manuscript.

4. Some spelling mistakes should be well corrected in line 92,164,267, etc.

Reviewer #4: The research presented in the work “Sex-specific and concentration-dependent influence of Cremophor RH 40 on ampicillin absorption via its effect on intestinal membrane transporters” is an interesting topic in relation to the most widely used excipient Cremophor RH 40 which is an important component of most pharmaceutical preparations.

The following are specific comments, questions and suggestions to improve the quality of the manuscript:

1. Since the focus of the manuscript was to find the effect of the widely-used pharmaceutical excipient Cremophor RH 40 on the intestinal absorption of ampicillin in male and female rats. Therefore, the title must express the true content of the paper and include the animal that the sex specific is related to, so I suggest addition the of the word (in rats) at the end of the title.

2. The introduction to the work was presented correctly, the only point is that no information was given regarding the excipient Cremophor RH 40, I suggest to add a small paragraph about this excipient and its physicochemical properties in the introduction part.

3. Why Cremophor RH 40 has been choose and not Cremophor EL as an excipient? did the second excipient also have the same effect on intestinal absorption of ampicillins?

4. The selection of methods is rather broad and cover the design-space well as were the reported research results.

5. the following used abbreviations should be defined:

P6 - L 101: PBS

P7 - L 127: LDH

P11- L 208: HEPES

P 12 - L 221: Tma-DPH

P 16 -L 325: CMC

Reviewer #5: The manuscript titled “Sex-specific and concentration-dependent influence of Cremophor RH 40 on ampicillin absorption via its effect on intestinal membrane transporters” reported the effects of Cremophor RH 40 on PepT1 protein recruitment on the membrane and led to the changes of ampicillin absorption in rats, which is interesting and new. The paper could even increase in quality by performing the following amendments:

1. Please explain the basis of the Cremophor RH 40 dosage used in the article and its relationship with the clinical preparations

2. transport experiments on Caco-2 cells are needed to verify the effects of Cremophor RH 40 on the PepT1 protein recruitment on the cell membrane

3. relevant references are missing in line 366

4. try to calculate/estimate the meaning of the interaction of Cremophor RH 40 and ampicillin observed in your experiments. Does the magnitude of interaction and doses reflect the clinical observations in literature? Or is it x-fold lower

6. PLOS authors have the option to publish the peer review history of their article (what does this mean?). If published, this will include your full peer review and any attached files.

Reviewer #1: **Yes: **Associate Professor/ Sally A. El-Zahaby, phD

Reviewer #2: **Yes: **Dr. Dhanusha Thambavita

Reviewer #3: No

Reviewer #4: **Yes: **Shahla S. Smail

Reviewer #5: No

---

## [Author Response · Author response to Decision Letter 0]

22 Jan 2022

Dear Professor Wang,

We wish to thank the reviewer for their insightful comments. 

We have amended the manuscript in accordance with the comments of the referee and provide attached a point-by-point response to the issues raised.

We have marked all the changes made in the revised manuscript.

Thank you for your time, and I look forward to hearing from you.

Yours sincerely,

Wenbin Deng

---

## [Editor Report · Decision Letter 1]

25 Jan 2022

Sex-specific and concentration-dependent influence of Cremophor RH 40 on ampicillin absorption via its effect on intestinal membrane transporters in rats

PONE-D-21-36367R1

Dear Dr. Deng,

We’re pleased to inform you that your manuscript has been judged scientifically suitable for publication and will be formally accepted for publication once it meets all outstanding technical requirements.

Kind regards,

Chun-Hua Wang

Academic Editor

PLOS ONE

---

## [Editor Report · Acceptance letter]

18 Feb 2022

PONE-D-21-36367R1 

Sex-specific and concentration-dependent influence of Cremophor RH 40 on ampicillin absorption via its effect on intestinal membrane transporters in rats 

Dear Dr. Deng:

I'm pleased to inform you that your manuscript has been deemed suitable for publication in PLOS ONE. Congratulations! Your manuscript is now with our production department. 

Kind regards, 

on behalf of

Dr. Chun-Hua Wang 

Academic Editor

PLOS ONE